# Overview of Key Techniques for In Situ Tests of Electromagnetic Radiation Emission Characteristics

**DOI:** 10.3390/s24237515

**Published:** 2024-11-25

**Authors:** Zhonghao Lu, Yan Chen, Yunxiao Xue

**Affiliations:** College of Electronic Science and Technology, National University of Defense Technology, Changsha 410073, China; chenyancy@nudt.edu.cn

**Keywords:** electromagnetic compatibility, in situ measurement, electromagnetic interference filtering, transient and broadband electromagnetic interference measurement, fast extraction

## Abstract

With the growing number of electronic devices loaded and increasing influence from electromagnetic interference, large-scale systems or platforms are confronted with increasingly severe electromagnetic compatibility challenges. Due to the vast size of these systems and the multitude of electronic devices they contain, standard laboratory environments are often inadequate for meeting test requirements. This paper reviews the state-of-art in the area of field measurement techniques related to the checking of electromagnetic compatibility, and the key technologies of electromagnetic interference filtering and wide-bandwidth, large-dynamic, and rapidly transient signal extraction in the measurement field are analyzed. The research status of electromagnetic interference suppression, transient and broadband measurement, and environmental interference suppression combined with time-domain fast measurement and other technologies are summarized and analyzed. Based on a comparative analysis of the aforementioned technologies, the future development trends of field measurement technology are also discussed.

## 1. Introduction

Electromagnetic compatibility (EMC) is the functional ability with which electronic equipment and systems can use the electromagnetic spectrum under the same electromagnetic environment to avoid the unacceptable degradation caused by electromagnetic radiation and sensitivity [1], thereby implementing their function [2]. Consequently, adhering to EMC standards is of paramount importance for all electronic devices and systems [3]. The modern information system is becoming increasingly complicated, as we are involved in the fast development of science and technology. Meanwhile, the number of pieces of electronic equipment loaded on the same platform is soaring, which also raises the requirement for EMC. On the other hand, with the electromagnetic environment becoming worse, electric products are confronted with increased electromagnetic interference (EMI), which leads to a significant increase in the risk of performance degradation or failure of the product itself. Therefore, research on EMC technology is both important and urgent.

Practice shows that research in EMC technology is based on a large number of tests and measurements, and measurement is an indispensable means in EMC research and system design [4]. As the senior statesman in the field of EMC, Professor C. R. Paul from the University of Kentucky in the United States, said, “For the final successful verification, perhaps there is no other field that relies so strongly on measurement as electromagnetic compatibility” [5]. At present, there are a lot of academic research studies on EMC measurement, but most of them focus on EMC measurement methods and operations in laboratories and standard measurement environments. These internationally recognized EMC measurement standards, established by organizations such as the FCC, ETSI, and IEC, are typically conducted in controlled testing environments such as shielded rooms or anechoic chambers to ensure the accuracy and reliability of the test results. As referenced in [6], according to the standards for radiated emissions (RE) tests in FCC Part 15 and CISPR 32, the RE compliance tests for many multi-modular devices operating at high frequencies (>10 GHz), such as router systems, are performed in a semi-anechoic chamber with a distance of 3 m. In reference [7], in order to ensure the accuracy of the test results of the radar system, according to the requirements of 4.3.2.1 Radio frequency (RF) absorber material and 4.3.4 ambient electrical level in MIL-STD-461G standard [8], during the experiment period, absorbing materials were laid between the radar main beam scanning range and the test platform, while ensuring that the test site was at the lowest level of the electromagnetic environment. Reference [9] mentions that typical networking equipment containing tens of nearly identical line cards and hundreds of optical modules, with radiation emission measurements performed in a commercial lab that meets the CISPR MIU (Measurement Instrumentation Uncertainty) requirements (measurement standards such as CISPR 16 [10] or IEC61000-4-21 [11] for the reverb chamber). Reference [12] shows that Equipment with wireless communication features such as Wi-Fi were tested in the laboratory according to ETSI EN 301 489-1 standards [13], but it was not very feasible to adjust the level of the equipment RF output power for measurement. The EMC measurement standard used by the U.S. Department of Defense for all equipment and systems, MIL-STD-461G, requires testing under specific conditions in the field environment, but a controlled laboratory environment is usually preferred to ensure the validity of the test results.

However, for large platforms, such as aircraft [14], ships [15], satellites [16], and high-speed rail [17,18], which are of large size and loaded with a large number of electronic devices, standard laboratory environments have difficulty in meeting the test requirements. For example, in the study of polymer materials, synchrotron radiation hard X-ray scattering characterizes the microstructure evolution law of polymer materials, and the microstructural changes of polymer materials in actual industrial processing can be truly reflected through a small synchrotron radiation in situ online research device [19]. For in situ gamma-ray radiation experiments, a simple two-panel device was tested for gamma-ray radiation in order to test the radiation survivability of the device platform. The experiments were carried out in a vacuum chamber with the device placed in front of a cobalt-60 source, which was switched on and off for a period of about 5 min while the potential difference between the cathode and anode was measured. This in situ test method can evaluate the impact of radiation on carbon nanotube field emission and physical structure and verify the performance of the device in the actual radiation environment [20]. In space radiation environment engineering, the ground simulation test equipment needs to be rationally arranged and monitored through as many channels as possible. In situ testing is required for macroscopic properties such as optical properties and electrical properties, as well as microscopic properties such as composition, structure, defects, morphology, etc. [21]. In the radiation reconnaissance of a nuclear accident site, the airborne radiation reconnaissance checks the passable route by flight and provides data support for the evacuation vehicle to choose the best route. This method simulates the real situation by placing different isotopes at different locations and creating measuring devices, and then planning further experiments to compare the results obtained from aerial reconnaissance with those obtained using established devices [22]. These cases show that in situ testing can effectively meet the standard requirements in the field environment in the measurement of radiation emission in large equipment, and the accuracy and reliability of the field test results are verified by comparing with the laboratory test results.

Therefore, it is necessary to research measuring and analyzing the electromagnetic radiation characteristics of these large devices or systems in their normal operating state in the in situ environment. This article primarily outlines the meaning and necessity of in situ electromagnetic radiated emission tests, technical challenges, and the research status of key technologies for measuring the electromagnetic radiation characteristics of large equipment or systems in their operational in situ environment.

## 2. The Meaning and Necessity of In Situ Electromagnetic Radiated Emission Tests

Electromagnetic compatibility measurement, which uses instruments to assess the EMC status of equipment and systems, is one of the major research areas in the field of EMC [23]. In situ system-level EMC testing refers to the measurement of large-scale equipment and platforms within their actual working electromagnetic environment, using specific technical means and methods to obtain the actual EMC status and accurately evaluate the EMC condition [24,25]. Measured parameters include the system safety factor, internal system electromagnetic compatibility, compatibility between systems, and electromagnetic interference between subsystems and equipment, among other aspects [26,27].

The necessities of in situ system-level EMC testing for large platforms such as aircraft, ships, and satellites can be summarized as follows. Firstly, there is no indoor standard site that can accommodate such large equipment under test (EUT), and it is impossible to transport the EUT to the standard open site for measurement. Secondly, because the system-level measurement and analysis of equipment require high-power radiation sources operating at full load, indoor measurement is obviously inappropriate. Moreover, most of the time, we must check and analyze these large platforms by in situ measurement to solve EMC problems in a timely manner, ensuring that the equipment and platform can successfully complete the various tasks assigned. Finally, the in situ environment is closer to the actual working environment of the EUT, and the measurement results can better reflect the actual situation.

The occurrence of EMI requires three essential elements: the EMI source, the coupling path, and the susceptible equipment. Harmonic and spurious emissions from radio electronic equipment on aircraft, ships, satellites, high-speed rail, and other large platforms are radiated by interconnecting cables or antennas, which can cause faults in sensitive equipment such as radio receivers due to wide bandwidth, high power, and other reasons. To address EMI issues, the first step is to clearly ‘see’ the electromagnetic radiation source, which can be achieved through in situ electromagnetic radiated emission testing.

As shown in Figure 1, the field measurement of electromagnetic radiation emission characteristics can quantify the electromagnetic radiation emission levels of various pieces of electromagnetic radiation equipment on large platforms under actual working conditions [28] and can provide accurate data support for analyzing and evaluating the EMC status of large platforms, as well as for investigating and resolving EMI issues.

## 3. Technical Challenges

### 3.1. Filtering the Electromagnetic Interference in the In Situ Measurement

Accurately extracting the emission characteristics of electromagnetic radiation from the EUT in a complex electromagnetic environment has always been a challenging issue, particularly when measurements are conducted in situ. As shown in Figure 2, various shortwave and ultrashort wave radio stations, television stations, and mobile communications, along with other potential environmental EMI sources in the vicinity of the measurement site, pose significant challenges to accurate measurement. The level of interference signals typically exceeds the EMC limit values by more than 30 dB [29], and the electromagnetic signals generated by the EUT may be modulated by environmental EMI. Therefore, when using a general EMI receiver to measure the radiated emission characteristics of the EUT in the field, the emission signal of the EUT is often masked by the environmental EMI levels. Consequently, this can lead to various adverse outcomes, such as erroneous measurement results, extended measurement times, and misidentifying environmental electromagnetic interference signals as those of the EUT, etc., making it impossible to accurately determine the electromagnetic radiation characteristics of the EUT. To accurately measure the electromagnetic radiation characteristics of the EUT in the field environment, it is essential to mitigate the impact of environmental EMI.

### 3.2. Fast Extraction of Wide-Bandwidth, Highly Dynamic, and Transient Signals

At present, the in situ testing of electromagnetic radiation emission characteristics faces the challenges of wide-bandwidth, highly dynamic, and transient signals. It is necessary to employ more advanced technical means in signal acquisition, real-time analysis, and signal characteristic presentation to minimize the possibility of missing hidden electromagnetic interference signals, especially transient signals, weak signals, and aliasing signals [30]. As shown in Figure 3, using general measurement tools such as EMI measurement receivers, sweep spectrum analyzers, and other measurement instruments based on the superheterodyne receiving system can lead to several drawbacks. Currently, the measurement method for electromagnetic radiation emission primarily adheres to the relevant provisions and requirements of EMC standard measurements. For instance, RE102 (10 kHz–18 GHz electric field radiated emission) and RE103 (10 kHz–40 GHz antenna harmonic and spurious output radiated emission) are specified in GJB152A ‘Electromagnetic Emission and Susceptibility Measurement of Military Equipment and Subsystems’. However, these standards only define the measurement methods and requirements for equipment or subsystems within a standard EMC measurement chamber, and they utilize automated detection instruments with swept-frequency spectrum analyzers. These standard methods are inadequate for filtering environmental EMI and extracting wide-bandwidth, highly dynamic, and transient interference signals in field measurements. Additionally, in China, there is only GJB1389A-2005 ‘System Electromagnetic Compatibility Requirements’ for the field measurements of EMC, which outlines the requirements and corresponding measurement methods for electric field intensity in real working environments. Consequently, the standard does not provide corresponding provisions for the measurement and quantification of electromagnetic radiation emission under field environmental conditions, failing to meet the needs for EMI signal detection and EMC problem resolution.

To assess the electromagnetic compatibility status of field electronic devices, it is necessary to detect the frequency, amplitude, and modulation parameters of their emitted signal in both the short and long term. Although the frequency-domain EMI test receiver can provide the accurate measurement of a large dynamic range, it also has great shortcomings in the in situ test. First, for the broadband signal, the test time is used to scan the frequency point, and the test equipment needs to be turned on for a long time, so the test greatly increases the test cost. For example, measurements in the 30 MHz to 1 GHz band usually take 30 min. If the residence time is set to 100 ms to improve the test accuracy, scanning the 200,000 frequency points will take more than 55 h. Secondly, the long time measurement increases the probability that the test accuracy is disturbed by other devices; again, the frequency-domain measurement usually cannot provide sufficient information, resulting in the loss of important information, and cannot describe the dynamic signal. Finally, the signals of the test equipment are often sudden, instantaneous, non-stationary signals, which must be captured quickly.

With the rapid development of modern signal processing technology, broadband time-domain test and analysis technology has great advantages in the field electromagnetic emission test. Time-domain field testing can extract a large amount of accurate information in a short measurement time. A digital processing method based on the Fourier transform can perform the in parallel processing of the amplitude and phase information of the signal and can access acquire frequency, amplitude, and modulation parameters. The measurement time can be shortened by at least one or even several orders of magnitude. The biggest difference between a time-domain EMI test system and a traditional EMI receiver is that it uses a real-time FFT processor to conduct the real-time time–frequency transformation of broadband signal and the capture of the signal by real-time triggering to obtain all kinds of information needed, avoiding the phenomenon that scanning EMI receivers often miss instantaneous events. Reference [31] refers to the radiation emission measurement of a carrier-borne equipment being conducted by using the time–frequency hybrid measurement receiver. It shows the spectral results measured in the time domain and frequency domain that are shown in Figure 4, and it shows the time-domain waveform of the emission signal of an equipment obtained by the field test shown in Figure 5. The time-domain observation time is 5 ms, and the frequency-domain sweep step value is 10 kHz. As can be seen from the figure, the amplitude spectra of the time-domain and frequency-domain measurements are basically matched. The frequency-domain measurement time needs 1500 ms, and the scan needs to be maintained in the maximum holding mode. After repeated scans, relatively stable results can be obtained.

## 4. The Research Status of Key Technologies

### 4.1. Research Status of Environmental Electromagnetic Interference Suppression Technology

The early environmental EMI suppression method is based on time-sharing measurement [32]. The specific operation process is as follows. Firstly, turn off the EUT at the moment $t_{1}$, and measure the power spectrum of the environmental electromagnetic signal $S_{t_{1}}\left(\omega \right)$ using EMI receiver or spectrometer; from this the power spectrum of the environmental EMI $P_{n(t_{1})}\left(\omega \right)$ can be obtained. Then, the EUT is turned on at the moment $t_{2}$, and the measured signal power spectrum is $S_{t_{2}}\left(\omega \right)$, which is composed of the ambient EMI power spectrum $P_{n(t_{2})}\left(\omega \right)$ and the EUT radiation signal power spectrum, that is, (Equation 1):(1)$$S_{t_{2}}\left(\omega \right)=P_{n(t_{2})}\left(\omega \right)+P_{EUT(t_{2})}\left(\omega \right)$$

Subtract the results of the two measurements (correlation processing) to eliminate the environmental EMI; then, the power spectrum of EUT radiation signal can be obtained as (Equation 2):(2)$$\begin{array}{cc} P_{EUT}\left(\omega \right) & =S_{t_{2}}\left(\omega \right)-S_{t_{1}}\left(\omega \right)\\ \; & =P_{n(t_{2})}\left(\omega \right)+P_{EUT(t_{2})}\left(\omega \right)-P_{n(t_{1})}\left(\omega \right)\\ \; & =P_{EUT(t_{2})}\left(\omega \right) \end{array}$$

Obviously, the precondition is that the power spectrum of the environmental EMI, measured twice, is almost equal whether the EUT is on or off. Thus, $P_{n(t_{2})}\left(\omega \right)\approx P_{n(t1)}\left(\omega \right)$, that is to say, the environmental electromagnetic interference signal is a time-stationary signal. In fact, in many cases, the interference signal in the ordinary environment is intermittent and amplitude fluctuates. For example, when there is a short-time burst signal with power spectrum $P_{shot\;time}\left(\omega \right)$, the ambient EMI signal with power spectrum $P_{n(t_{1})}\left(\omega \right)$ measured when the EUT is turned off and the ambient EMI signal with power spectrum $P_{n(t_{2})}\left(\omega \right)$ measured when the EUT is turned on cannot be effectively offset. In addition, in many field measurement cases, the equipment is not allowed to be turned on and off. For example, when measuring the mobile base station, turning off the base station will lead to the interruption of signal transmission and reception, resulting in the failure to provide services for end users and other worse consequences.

With the proposal of the adaptive noise cancellation principle [33], various methods of environmental EMI suppression have been proposed. Reference [34] describes the application of adaptive noise cancellation to the measurement of the electromagnetic environment in urban areas. American scholar M. A. Marino Jr. first proposed the concept of virtual darkroom and obtained the patent technology of virtual darkroom in 2005 [35]. Based on the principle of adaptive noise cancellation, the patent overcomes the time difference problem caused by single-channel time-sharing measurement by means of dual-channel reception, and filters out the external electromagnetic radiation not belonging to EUT, which can improve the measurement accuracy and expand the application range. The U.S. military has purchased a virtual darkroom system for EMI measurement, CASSPER, which is designed and implemented according to the patent.

The system diagram of CASSPER is shown as Figure 6, which utilizes two channels that are synchronized in time and frequency to receive signals simultaneously [36]. The function of the high-speed digital signal processor is to calculate the correlation of time, frequency, and phase to filter the environmental EMI. A signal $S_{d}\left(\omega \right)$ from a mixture of the EUT and ambient EMI is received at a distance of d in front of the EUT (marked as A). At the same time, the EUT interference signal $S_{10d}\left(\omega \right)$ is received at a distance of 10d or further (marked as B). If the environmental electromagnetic interference at the measurement site is stationary, the EUT interference signal received at B is at least 20 dB lower than that received at A. Moreover, the environmental EMI values measured at two positions are basically the same. On the premise of this, adaptive noise cancellation can be applied. Obviously, the signal spectrum of EUT is the difference between the signal spectra at A and B, as in (Equation 3):(3)$$\begin{array}{cc} P_{EUT}\left(\omega \right) & =S_{d}\left(\omega \right)-S_{10d}\left(\omega \right)\\ \; & =\left(P_{n(d)}\left(\omega \right)+P_{EUT(d)}\left(\omega \right)\right)-(P_{n(10d)}\left(\omega \right)+\\ \; & \;\;P_{EUT(10d)}\left(\omega \right))\\ \; & \approx \left[1-1/\left(100d^{2}\right)\right]\cdot P_{EUT(d)}\left(\omega \right)\\ \; & \approx P_{EUT(d)}\left(\omega \right),\left(P_{n(d)}\left(\omega \right)\approx P_{n(10d)}\left(\omega \right)\right) \end{array}$$

The radiation signal spectrum of an EUT recovered by CASSPER is shown in Figure 7.

### 4.2. Research Status of Transient and Broadband EMI Measurement Technology

It should be emphasized that the above measurement methods are based on frequency sweep, which takes a long time and may miss burst signals when operating within a wide bandwidth. Currently, the field measurements of radiated emission often confront the challenges of wide-bandwidth, highly dynamic, and transient burst signals. Therefore, it is necessary to implement effective measures in signal acquisition, real-time analysis, and signal feature extraction to minimize the possibility of missing significant signals, especially transient, weak, and aliasing signals.

Russer, a professor from the Technical University of Munich, Germany, and his team [37] proposed the conception of time-domain EMI measurement [38,39]. The measurement system is triggered in real time according to the information of the signal. It uses high-speed data acquisition technology and digital down conversion technology to realize pure digital intermediate frequency signal and a real-time fast Fourier transform (FFT) processor to analyze the broadband digital signal in the time and frequency domains. It can measure not only periodic and random signals, but also instantaneous signals by keeping the time relationship between signals, and can realize various signal analyses. Compared with the traditional EMI receiver, the time-domain EMI measurement system uses real-time trigger to capture the signal and real-time FFT processor to transform the broadband signal in the time–frequency domain, which can obtain all kinds of required information and avoid missing the instantaneous signal. Prof. Russel and his team have continuously improved and perfected the time-domain EMI measurement receiver and given the evaluation criteria for the measurement uncertainty in the time-domain EMI measurement receiver [40]. Finally, an electromagnetic interference time-domain measurement system is formed, which is in line with CISPR 16-1-1standard [41]. The team founded GUASS Instrument and introduced the TDEMI-X, an EMI time-domain measurement receiver.

In June 2010, IEC published amendment 1 to CISPR 16-1-1, 3rd edition, which allows the introduction of FFT-based measurement instruments in EMI measurements. The release of this standard provides a standard and legal basis for time-domain EMI measurement receivers [42]. As a result, the major instrument manufacturers in the electronic measurement industry have also introduced time-domain EMI interference measurement receivers. For example, Tektronix has introduced the RSA3000B series of real-time spectrum analyzers that provide the real-time RF viewing of the spectrum for a variety of digital RF applications, including radio frequency identification (RFID), radio management, and spectrum management. The RSA3000B adopting Digital Phosphor technology (DPX) can process more than 48,000 spectrum measurements per second, which is much higher than other spectrum analyzers without DPX. The DPX technology works on the following principle: the analog RF input signal is first down-converted and sampled, and then converted into the frequency domain using a discrete Fourier transform. These spectral data are sent to the DPX pixel storage buffer, accumulating into a histogram of the pixel occurrence. The DPX display processor processes the measurement data and output them to the display at a speed of about 10 refresh per second, so that short events can be kept on the display long enough for human observation; the color grading is used to indicate the frequency and probability of signal occurrence, so that the user can easily identify the characteristics of different signals. With the Frequency Mask Template (FMT) function, one can find interference signals and transient signals that cannot be found by other instruments. By displaying a seamless record of frequency and power changes over time, Tektronix’s analyzers can spot many transient signal problems, such as modulation switching in software-defined radio systems, identifying problem pulses in radar transmissions, determining dynamic modulation changes during WLAN transmissions, and so on. Rohde and Schwarz have developed the R&S FSVR real-time spectrum analyzer, which combines the functions of signal analysis, spectrum measurement, and real-time spectrum analysis. Specifically, R&S FSVR, in real-time mode, calculates up to 250,000 spectra per second by capturing the RF signal with a bandwidth of up to 40 MHz and converting it to the spectrum. These spectrum data are FFT transformed to convert the time-domain signal into the frequency-domain signal to realize the spectrum analysis of the signal. Furthermore, digital signal processing technology is used to digitally filter, balance, and correct spectral data to improve the accuracy and reliability of spectral analysis. This combination of FFT and digital signal processing technology enables R&S FSVR to monitor and analyze broadband signals in real time, providing high-precision and high-resolution spectral analysis results. The R&S FSVR can detect various sporadic and ultrashort transient signals. FSVR can provide all functions of signal analysis and spectrum measurement for both civil communication systems, such as LTE, WiMAX, WLAN, Bluetooth, and RFID, and military systems, such as radar and frequency-hopping communication. The measurement speed of FSVR is more than 5 times faster than that of similar spectrometers. Keysight has introduced the X series of signal analyzers, which offer a measurement frequency range of up to 50 GHz, integrated 1 GHz signal analysis bandwidth, and ultra-wide real-time data stream transmission capabilities, enabling the better analysis of transient signals and signals that are difficult to capture. It has the functions of real-time spectrum analysis, frequency template trigger, time-limited trigger, and complete vector signal analysis.

### 4.3. Research Status of Environmental Electromagnetic Interference Suppression Combined with Time Domain Fast Measurement Technology

Although these new time-domain EMI measurement receivers have greatly improved the measurement speed, they basically adopt a single-channel structure and have limited ability to filter the EMI in the field environment. To this end, GUASS Instrument launched its high-end product, the TDEMI-X time-domain measurement receiver, in 2013 through continuous research on time-domain electromagnetic interference field measurement systems [43,44].

The instrument can complete the rapid measurement from 10 Hz to 40 GHz at a rate 64,000 times faster than other similar products and has the real-time analysis capability from 325 MHz to 40 GHz [45]. The most attractive aspect is that the system adopts a dual-channel mode to introduce an adaptive noise cancellation system to eliminate the environmental EMI in the field measurement. However, according to the research results by the University of Electronic Science and Technology of China [46] and the Ordnance Engineering College [47], the virtual darkroom measurement method based on adaptive noise cancellation is not ideal for in situ measurement. This is because, in order to prevent the reference channel from receiving the EUT signal, the two receiving antennas in the dual channel are far away from each other, resulting in inconsistency in the environmental EMI between the two locations. However, reducing the distance between antennas will make the EUT signal leak to the reference channel, resulting in increased signal correlation and reduced interference suppression effect. When there is a strong component with the same frequency as the EUT radiation signal in the environment, the interference suppression effect is not obvious. In fact, the interference cancellation structure based on beamforming [48] still has good interference suppression effect without a reference channel.

Therefore, in 2012, we proposed a multi-channel measurement system based on the principle of spatial filtering [49,50], which has the effect of environmental EMI suppression and does not require prior information and reference information [51,52]. Multiple Signal Classification (MUSIC) was used for Direction of Arrival (DOA) estimation, and MVDR beamforming was used for optimal weight calculation. The combination of short-time Fourier transform (STFT) and the minimum variance distortion-free response (MVDR) algorithm for dynamic time–frequency analysis is proposed, and a method for suppressing environmental interference in electromagnetic compatibility testing is proposed. The experimental results are shown in Figure 8, and the narrowband interference signal is suppressed by at least 57.8 dB at 420 MHz. For the pulse-modulated signal, the experimental results show that the interfering signal is suppressed to the noise level while maintaining the characteristics of the measured signal with almost no distortion. While environmental EMI mitigation methods based on multi-channel systems are effective, they require precise calibration and synchronization to ensure the accuracy and consistency of multi-channel systems, which may increase the challenges of system deployment and maintenance, resulting in difficult and costly implementation, limiting their wide range of applications.

Reference [53] refers to the method of synthetic aperture and performs beam synthesis after receiving signals at each position by moving antennas. The prerequisite for this method is still that the EUT radiation signal and interference signal are stable signals in time. Reference [54] enhanced the configuration of the measurement antenna by housing it within a metallic cavity with a single open side to shield against environmental interference signals from other directions. However, this approach has a drawback: when both the interference and the EUT’s radiated signals fall within the antenna’s effective radiation pattern, environmental electromagnetic interference cannot be effectively filtered out. Additionally, the cavity’s resonant effects can lead to a decrease in the accuracy of the measurement outcomes. It is worth noting that references [55,56] introduced the method of time–frequency joint multi-domain analysis based on the system of time-domain signal reception. It extracts the time–frequency characteristics of environmental EMI by wavelet transformation and filters or attenuates the interference by the given threshold function to obtain the actual electromagnetic radiation signal of EUT.

References [57,58] proposed a multi-channel time-domain rapid measurement and signal calculation method for low-frequency electromagnetic interference and a multi-channel time-domain rapid measurement system for low-frequency electromagnetic interference is developed and can only measure low-frequency electromagnetic interference of current, voltage, magnetic field and electric field within 100 MHz. Its basic principle is to use measurement sensors to extract complex electromagnetic environment and low frequency electromagnetic interference signal in large system, after low-pass filtering, and then converted into digital signal by multi-channel time domain acquisition card and FFT calculation, and then transmitted to the computer for data reprocessing. This method can be measured simultaneously in multiple channels, and adopts the time-domain method, with high measurement efficiency. As can be seen from Figure 9, the standard signal source outputs a signal of 106.98 dBμV, and the multi-channel time-domain low-frequency electromagnetic interference measurement system can simultaneously complete the electromagnetic signal measurement of four channels. The maximum error of the full band in the four channels is only 0.38 dB, which can greatly improve the measurement efficiency. The error between the modified calculation result and the standard measurement method in the frequency domain is less than 1.0 dB. However, limited by the sampling rate, the data post-processing ability, and the sensitivity of the current oscilloscope, it is only suitable for the low-frequency field.

Reference [59] proposed a virtual chamber measurement method based on a spatial domain cancellation technique for radiation emission in situ test. According to this method, the authors designed a dual-channel in-site measurement system of electromagnetic radiation characteristics based on spatial filtering and time-domain signal acquisition [60], which can detect the electromagnetic radiation performance of large equipment and platforms, mainly in the frequency range of 300 MHz to 6 GHz. The binary broadband antenna array is used to receive the signal and the signal acquisition and processing module operates first through the RF receiver, then through the low noise amplifier and mixer, and then through the digital signal of analog-to-digital converter (ADC). Then, the digital signal processing algorithm in the module is performed using the field-programmable gate array (FPGA). Finally, connect to the software operating system for real-time signal processing. The basic principle of this system is shown in Figure 10. Using the MVDR beamforming method, the weighted value is calculated, and the array pattern is synthesized so that the zero trap of the pattern is aligned with the direction of the EUT signal, so as to extract the spectrum of the interference signal and then obtain the real radiation characteristics of the device under test through spatial filtering and cancellation techniques. The experimental results show that the proposed method is effective for both broadband and narrowband interference, which can significantly suppress pulse modulation interference and ensure that the radiation characteristics of the device under test are not distorted, and that multipath interference is also effectively suppressed. Experimental data in Figure 11 show that LFM interference is suppressed by about 27 dB. While the prior information such as the number and azimuth distribution of electromagnetic interference sources in the test site environment is not needed, the system can suppress all types of interference at the same time and adapt to high-power equipment and has a wide measurement range and a fast time-domain measurement speed. Its interference suppression ability exceeds 20 dB, and the cost is low.

## 5. Conclusions

By comparing the domestic and abroad research status, some conclusions can be obtained. Firstly, the method based on dual-channel adaptive noise cancellation is commonly used to solve the problem of background noise in the field measurement of electromagnetic radiation emission. However, there are some limitations in practical application, such as insufficient environmental adaptability and limited anti-interference ability. Secondly, the measurement system of environmental EMI suppression based on multi-channel array signal processing is complex, the design and implementation is difficult, and the hardware development cost is high, which limit its wide application. Furthermore, the multi-channel array processing method requires the strict consistency of each channel which relies on complex calibration algorithm and high-cost hardware. Thirdly, the receiving system of time-domain EMI measurement has the ability to capture and extract non-stationary signals and transient burst signals, but has no ability to suppress EMI in the field environment. Nevertheless, the dual-channel in situ measurement system of electromagnetic radiation characteristics based on spatial filtering and time-domain signal acquisition, without the prior information of the number of environmental electromagnetic interference sources and the azimuth distribution, can still create a dark room environment equivalent to filter out the measurement of environmental electromagnetic interference and obtain the actual radiation characteristics of the EUT.

In the future, the direct acquisition technology of ultra-wideband RF signals can be studied, and the analysis of the collected signals in the time domain, probability domain, and power domain can be further explored. A more effective spatial spectral estimation and beamforming algorithm will be designed to reduce the amount of calculation and improve test speed, achieving the accurate extraction of EUT electromagnetic radiation characteristics in in situ measurements, as well as the measurement and filtering of environmental interference signals. Concurrently, with the advancement of artificial intelligence technology, machine learning algorithms will be integrated to automatically identify and classify types of electromagnetic interference and extract useful signals.

## 6. Patents

Lu, Z.-H., Zhou, D.-M., Li, G.-S., Qin, Y.-J., Huang, J.-J., Liu, J.-B., Xue, G.-Y., and Liu, P.-G. (2015). “An Innovative In-Situ Equivalent Virtual Chamber Measurement Method”, CN 201510420443.0, Application No. 201510420443.0, Application date: 16 July 2015 [60].

## Figures and Tables

**Figure 1 sensors-24-07515-f001:**
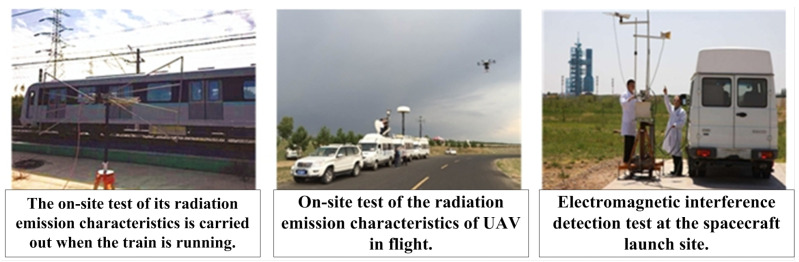
Typical application scenarios for the in situ measurement of electromagnetic radiation emission characteristics.

**Figure 2 sensors-24-07515-f002:**
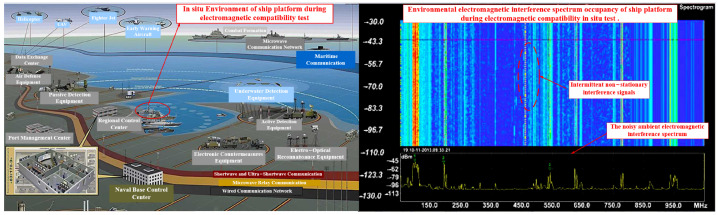
Ambient electromagnetic interference and its emission characteristics in the in situ measurement.

**Figure 3 sensors-24-07515-f003:**
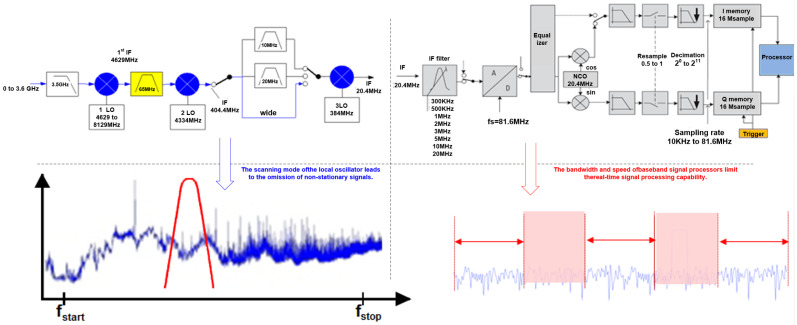
Drawbacks of swept EMI receiver.

**Figure 4 sensors-24-07515-f004:**
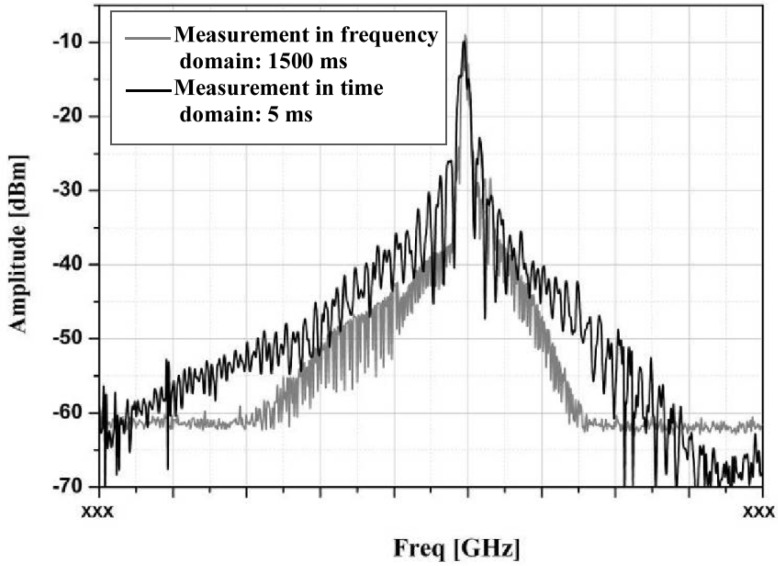
Spectral results of time-domain measurement and frequency-domain measurement at different observation times.

**Figure 5 sensors-24-07515-f005:**
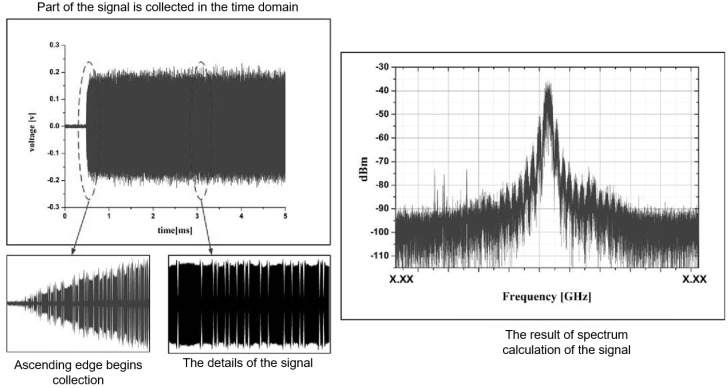
Non-stationary EMI signal acquisition results.

**Figure 6 sensors-24-07515-f006:**
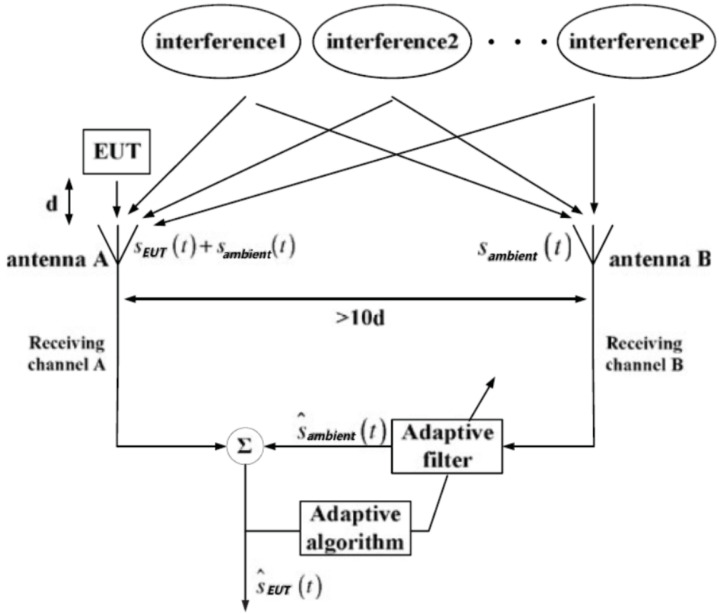
Structure diagram of the CASSPER system.

**Figure 7 sensors-24-07515-f007:**
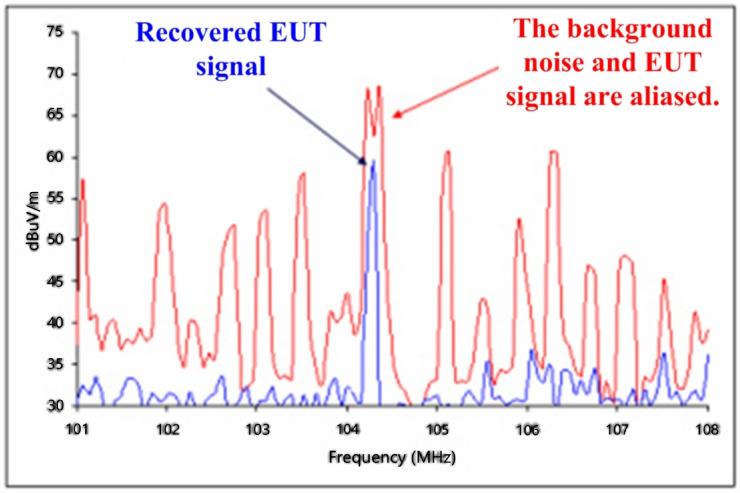
Test results of the CASSPER system.

**Figure 8 sensors-24-07515-f008:**
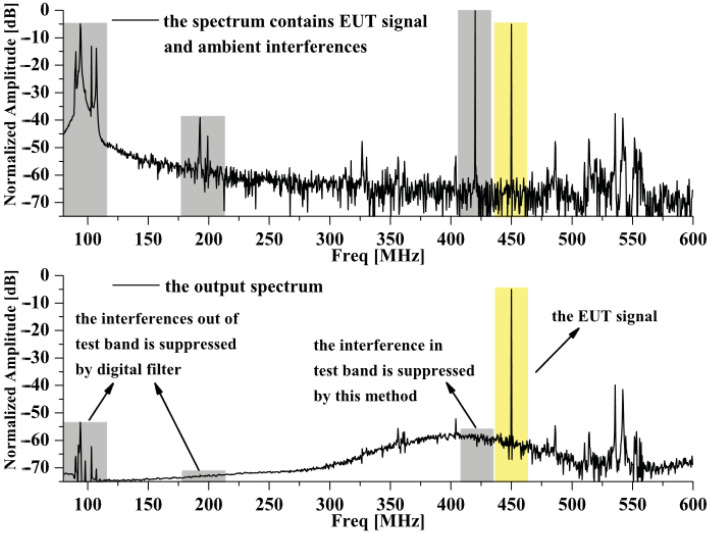
Suppression effect on narrowband interference.

**Figure 9 sensors-24-07515-f009:**
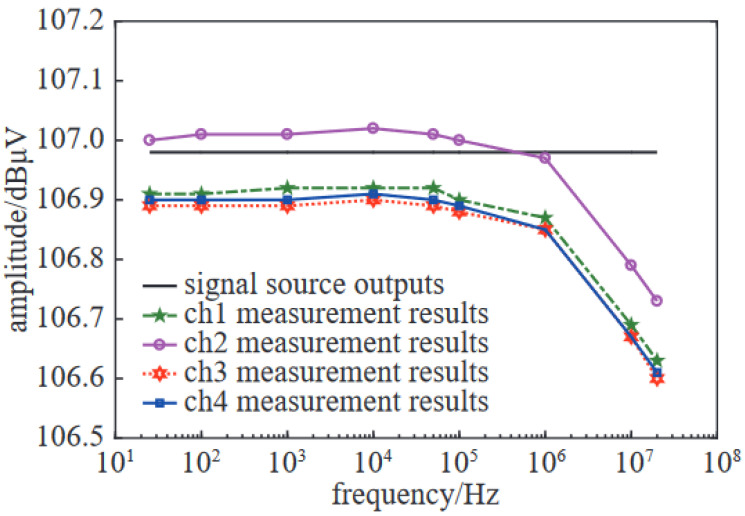
Verification results of the electromagnetic interference multi-channel time domain rapid measurement system.

**Figure 10 sensors-24-07515-f010:**
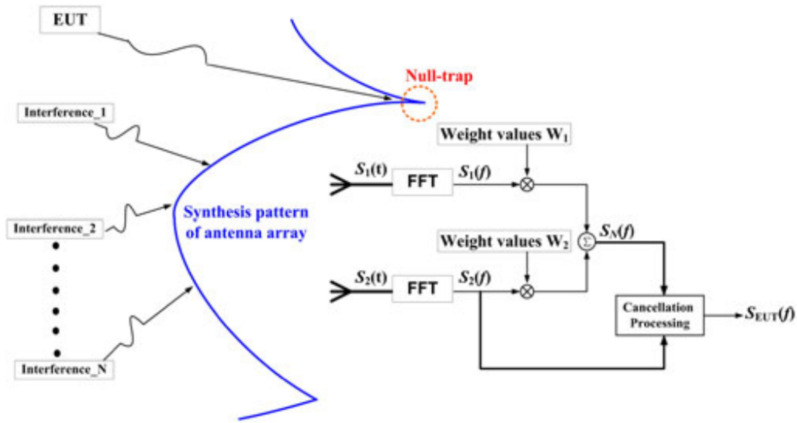
Schematic for the equivalent virtual chamber measuring method based on spatial cancellation.

**Figure 11 sensors-24-07515-f011:**
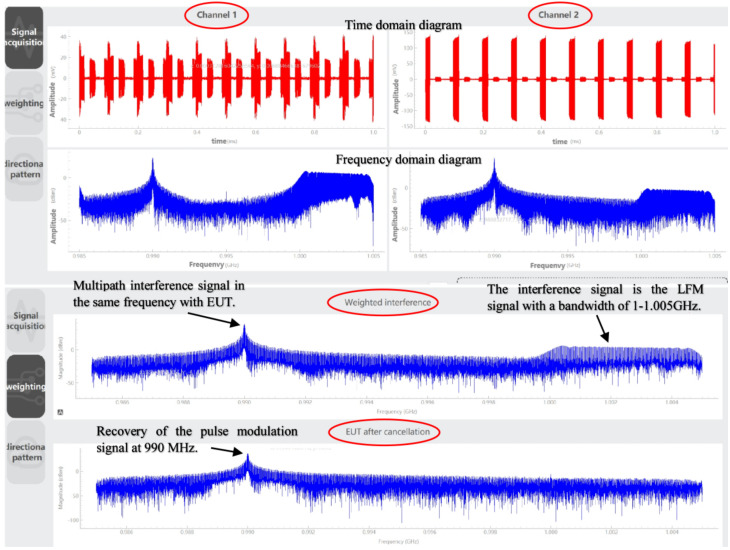
Test results of the electromagnetic radiation characteristic measurement system based on the spatial filtering and time-domain signal acquisition.

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
