# Peer review of "Overview of Key Techniques for In Situ Tests of Electromagnetic Radiation Emission Characteristics"

_sensors, 2024, doi:10.3390/s24237515_

Round 1

Reviewer 1 Report

Comments and Suggestions for Authors The manuscript reviews the state-of-art in the area of field measurement techniques related to the check of electromagnetic compatibility. The purpose of the measurements is to detect, to measure, and to characterize spurious microwave radiation from the objects like trains, planes, ships, etc., which are too large to be placed in an anechoic chamber (dark room according to author’s terminology). Unfortunately, the radiation magnitude under measurement is typically comparable to the background electromagnetic radiation at the measurement site originating from local TV stations, mobile base stations, car ignition systems, etc. In addition, the spurious radiation may be pulse-like and unstable in time. Therefore, such measurements involve a number of sources of measurement uncertainty that must be reduced. For this purpose, some sophisticated techniques are suggested in the literature. These techniques are described, analyzed, and compared in the manuscript. Presently, this area attracts much attention and is fast-growing. Therefore, this review is timely and may be of interest for researchers working in this area. The manuscript contains much useful information (at least, from my point of view). The consideration seems to be thorough, and the style of writing is solid enough, except for few misprints. The references and figures are appropriate.

My comments to the authors are following.   1. Please consider adding few words in the abstract to emphasis clearly the purpose of the manuscript, such as: "The paper reviews the current state-of-art in the field" or something of this kind. This would help a reader to search necessary information in the publication databases.  2. Line 225: "Russer. P and his team..." - "Prof. Russel and his team" would be more appropriate. The same in line 196: why "Marino. J"? If you wish to include initials, they must be placed before th family name. Also, according the reference list, the initials are different. 3. Some editing may be needed: line 19: why "electromagnetism compatibility", not "electromagnetic"? Line 252 - the sentence is started with "And"; line 205: the sentence is started with abbreviation "Fig."

Author Response

Dear Reviewer 1:

A revised version of our manuscript ‘Overview of Key Techniques for In-situ Test of Electromagnetic Radiation Emission Characteristics’ is attached in this mail. We would like to resubmit it as a research paper to be published in Sensors.

 The comments from you and the other reviews were highly insightful and enabled us to greatly improve the quality of our manuscript. The item-by-item responses to each of the comments of the reviews are included at the bottom of this letter.

 The literature has been revised carefully according to the comments. We hope that the revision in the manuscript and our accompanying responses will be sufficient to make our manuscript suitable for publication.

 Once again, thank you and the other reviewers for your meticulous work and your helpful comments on our manuscript. We look forward to hearing from you at your earliest convenience. 

Yours sincerely,

Zhong-hao Lu, Yun-xiao Xue

Address:College of electronic science and technology, Nation University of Defense Technology,Changsha, 410073, China

Tel:+8607318457445
E-mails: luzhonghao@nudt.edu.cn、xueyunxiao@nudt.edu.cn

For research articleOverview of Key Techniques for In-situ Test of Electromagnetic Radiation Emission Characteristics

Manuscript ID: sensors-3268399

Response to Reviewer 1 Comments

1. Summary

Once again, thank you and the other reviewers for your meticulous work and your helpful comments on our manuscript.

2. Point-by-point response to Comments and Suggestions for Authors

Comments 1: Please consider adding few words in the abstract to emphasis clearly the purpose of the manuscript, such as: "The paper reviews the current state-of-art in the field" or something of this kind.  This would help a reader to search necessary information in the publication databases. 

Response 1: Thank you for pointing this out. We agree with this comment. Therefore, we have changed the abstract. Page 2, Line 84 of the revised manuscript reads: The paper reviews the state-of-art in the area of field measurement techniques related to the check of electromagnetic compatibility.

Comments 2: Line 225: "Russer.  P and his team..."  - "Prof. Russel and his team" would be more appropriate.  The same in line 196: why "Marino.  J"?  If you wish to include initials, they must be placed before th family name.  Also, according the reference list, the initials are different.

Response 2: Thank you for pointing this out. We agree with this comment and have changed:

Line 213 (Original line 225) : Prof. Russel and his team

Line 174 (Original line 196): American scholar M. A. Marino Jr.

[26]M. A. Marino Jr., “System and method for measuring RF radiated emissions in the presence of strong ambient signals,” U.S. Patent 6980611B1, 2005.

Comments 3: Some editing may be needed: line 19: why "electromagnetism compatibility", not "electromagnetic"?  Line 252 - the sentence is started with "And";  line 205: the sentence is started with abbreviation "Fig."

Response 3: Agree. We have revised:

line 20 (Original line 19):  What we mean is that adhering to EMC standards is of paramount importance for all electronic devices and systems. The sentence has been corrected.

Line 240 (Original line 252): Deleted "and", the sentence is started with " The measurement speed of FSVR".

Line 193 (Original line 205): The radiation signal spectrum of an EUT recovered by CASSPER is shown in Fig.\ref{Fig.5}.

3. Response to Comments on the Quality of English Language

Point 1: The consideration seems to be thorough, and the style of writing is solid enough, except for few misprints.

Response 1: We have meticulously reviewed the manuscript in English, ensuring that the completeness of the content is preserved, while also endeavoring to enhance the text's fluency and readability without deviating from the core viewpoints.

4. Additional clarifications

The literature has been revised carefully according to the comments.We hope that the revision in the manuscript and our accompanying responses will be sufficient to make our manuscript suitable for publication.

Reviewer 2 Report

Comments and Suggestions for Authors

This manuscript provides an overview of measurement techniques for on-site electromagnetic radiation emission characteristics in the EMC field, with a particular focus on ambient interference suppression and transient measurements. It introduces the challenges associated with on-site testing and fast signal processing for large, high-cost EUTs, and reviews state-of-the-art technologies such as adaptive noise cancellation, multi-channel spatial filtering, and fast time-domain signal processing. However, the structure of the article is rather disorganized, the writing lacks polish, and the format is not standardized. I don’t think it currently meets the requirements of a journal article. Major revisions are recommended before resubmission. Specific comments are as follows:

  1. The Introduction section spends too much space discussing the importance of EMC and related testing methods, only introducing the issue of large EUT on-site testing in the final paragraph. The focus is unclear, and various standards and numbers are presented in a chaotic format, significantly reducing readability.
  2. The necessity of fast signal processing is not clearly explained. There is no comparison of the processing speed of various methods against time requirements, and the time-domain measurement method is introduced without sufficient justification.
  3. The second research section lists many equipment models from commercial manufacturers, but lacks specific technical details and representative key technologies, which are not closely related to the content before and after this section.
  4. The resolution of the images is very low, especially Figures 2 and 3, where the text is almost illegible.
  5. Inappropriate descriptions are found throughout the article. For example:
  • There are some inconsistent statements. Is it "in-situ" or "on-site" testing?
  • Page 3, Line 92: "when EMC problems arise, we must check and analyze them by on-site measurement to solve them in time." I think EMC testing does not require immediate solutions.
  • Page 3, Line 97: "EMI is formed by three elements." In fact, coupling paths and sensitive equipment are not elements of EMI.
  • Page 7, Line 239: "digital fluorescence technology (DPX)" is incorrect. Actually DPX stands for Digital Phosphor eXpress.
  • Page 12, Line 360: "It has been studied for many years, but it has not been proved to be the best method." This conclusion is inappropriate and does not align with the theme of key techniques.
  • Page 12, Line 361: "Secondly, the measurement system of environmental EMI suppression based on multi-channel array signal processing needs to know the number of interference sources in advance." This point was not discussed earlier in the text. Conclusions should summarize previous content, not introduce new results.
Comments on the Quality of English Language

Many terms are used inaccurately or incorrectly, such as:

  • Page 2, Line 84: "inter-system electromagnetic compatibility" is repeated twice.
  • Page 5, Line 167: "and the measured signal power spectrum is..."
  • Page 7, Line 210: "large bandwidth"
  • Page 7, Line 229: "launched the receiver"
  • Page 9, Line 302: "...the antenna in a metal cavity with only one side opened."
  • Page 12, Line 370: "airspace filtering" should be "spatial filtering."
  • Page 13, Line 384: "6. Patents"

Author Response

Dear Reviewer 2:

A revised version of our manuscript ‘Overview of Key Techniques for In-situ Test of Electromagnetic Radiation Emission Characteristics’ is attached in this mail. We would like to resubmit it as a research paper to be published in Sensors.

 The comments from you and the other reviews were highly insightful and enabled us to greatly improve the quality of our manuscript. The item-by-item responses to each of the comments of the reviews are included at the bottom of this letter.

 The literature has been revised carefully according to the comments.We hope that the revision in the manuscript and our accompanying responses will be sufficient to make our manuscript suitable for publication.

 Once again, thank you and the other reviewers for your meticulous work and your helpful comments on our manuscript. We look forward to hearing from you at your earliest convenience. 

Yours sincerely,

Zhong-hao Lu, Yun-xiao Xue

Address:College of electronic science and technology, Nation University of Defense Technology,Changsha, 410073, China

Tel:+8607318457445
E-mails: luzhonghao@nudt.edu.cn、xueyunxiao@nudt.edu.cn

For research articleOverview of Key Techniques for In-situ Test of Electromagnetic Radiation Emission Characteristics

Manuscript ID: sensors-3268399

Response to Reviewer 2 Comments

1. Summary

Once again, thank you and the other reviewers for your meticulous work and your helpful comments on our manuscript.

2. Point-by-point response to Comments and Suggestions for Authors

Comments 1: There are some inconsistent statements. Is it "in-situ" or "on-site" testing?

Response 1: Thank you for pointing this out. "In-situ testing" emphasizes that the testing or measurement is conducted at the original location of the sample or system, without relocating the sample to another place. This type of testing helps to preserve the sample's pristine condition, preventing any changes that might occur during transportation. "On-site testing," on the other hand, focuses more on conducting tests at a specific physical location. Therefore, we have changed the manuscript to uniformly use "in-situ testing" to align with the theme of this paper.

Comments 2: Page 3, Line 92: "when EMC problems arise, we must check and analyze them by on-site measurement to solve them in time." I think EMC testing does not require immediate solutions.

Response 2: Thank you for pointing this out. What we mean to convey is that,, in most cases, timely measurement is required.

Page 2, Line 80 (Original Page 3, line 92): Most of the time, we must check and analyze these large platforms by in-situ measurement to solve EMC problems in a timely manner, ensuring that the equipment and platform can successfully complete the various tasks assigned.

Comments 3: Page 3, Line 97: "EMI is formed by three elements." In fact, coupling paths and sensitive equipment are not elements of EMI.

Response 3: Thank you for pointing this out. What we mean to convey is that the occurrence of EMI requires three essential elements: the EMI source, the coupling pathway, and the susceptible equipment. This highlights the three key components involved in the phenomenon of EMI. When discussing EMI, these three elements are typically considered because they collectively constitute the overall scenario of an EMI issue: without a source, EMI would not be generated; without a coupling pathway, EMI energy could not propagate to other locations; and without susceptible equipment, the effects of EMI would not be manifested.

Page 2, line 86 (Original Page 3, line 97):  The occurrence of EMI requires three essential elements: the EMI source, the coupling path, and the susceptible equipment.

Comments 4: Page 7, Line 239: "digital fluorescence technology (DPX)" is incorrect. Actually DPX stands for Digital Phosphor eXpress.

Response 4: Thank you for pointing this out. Through our research, we have learned that "DPX" is a registered trademark of Tektronix Corporation to describe its unique digital phosphor technology. Tektronix patented Digital Phosphor technology, or DPX®, is used in Real-Time Spectrum Analyzers (RSAs) to reveal signal details that are completely missed by conventional spectrum analyzers and vector signal analyzers.

Page 7, Line 227 (Original Page 7, Line 239): Digital Phosphor technology(DPX)

Comments 5: Page 12, Line 360: "It has been studied for many years, but it has not been proved to be the best method." This conclusion is inappropriate and does not align with the theme of key techniques.

Page 12, Line 361: "Secondly, the measurement system of environmental EMI suppression based on multi-channel array signal processing needs to know the number of interference sources in advance." This point was not discussed earlier in the text. Conclusions should summarize previous content, not introduce new results.

Response 5: Thank you for pointing this out. We have revised the paper:

Page 12, Line 348 (Original Page 12, Line 360): However, there are some limitations in practical application, such as insufficient environmental adaptability and limited anti-interference ability.

Page 12, Line 349 (Original Page 12, Line 361): Secondly, the measurement system of environmental EMI suppression based on multi-channel array signal processing is complex, the design and implementation is difficult, and the hardware development cost is high, which limits its wide application.

3. Response to Comments on the Quality of English Language

Point 1:

Page 2, Line 84: "inter-system electromagnetic compatibility" is repeated twice.

Page 5, Line 167: "and the measured signal power spectrum is..."

Page 7, Line 210: "large bandwidth"

Page 7, Line 229: "launched the receiver"

Page 9, Line 302: "...the antenna in a metal cavity with only one side opened."

Page 12, Line 370: "airspace filtering" should be "spatial filtering."

Page 13, Line 384: "6. Patents"

Response 1: We have meticulously reviewed the manuscript in English, ensuring that the completeness of the content is preserved, while also endeavoring to enhance the text's fluency and readability without deviating from the core viewpoints.

Page 2, Line 72 (Original Page 2, Line 84): Measured parameters include the system safety factor, internal system electromagnetic compatibility, compatibility between systems, and electromagnetic interference between subsystems and equipment, among other aspects.

Page 5, Line 155 (Original Page 5, Line 167): and the measured signal power spectrum is $S_{t_2}\left(\omega\right)$

Page 7, Line 198 (Original Page 7, Line 210): wide bandwidth

Page 7, Line 217 (Original Page 7, Line 229): The team founded GUASS Instrument and introduced the TDEMI-X, an EMI time domain measurement receiver.

Page 9, Line 290 (Original Page 9, Line 302): Reference \cite{b44} enhanced the configuration of the measurement antenna by housing it within a metallic cavity with a single open side to shield against environmental interference signals from other directions.

Page 12, Line 358 (Original Page 12, Line 370): spatial filtering

Page 13, Line 372 (Original Page 13, Line 384): 6. Patents

Lu, Z.-H., Zhou, D.-M., Li, G.-S., Qin, Y.-J., Huang, J.-J., Liu, J.-B., Xue, G.-Y., and Liu, P.-G. (2015). "An Innovative In-Situ Equivalent Virtual Chamber Measurement Method," CN 201510420443.0, Application No. 201510420443.0, Application date: Jul. 16, 2015.

4. Additional clarifications

Comments : The Introduction section spends too much space discussing the importance of EMC and related testing methods, only introducing the issue of large EUT on-site testing in the final paragraph. The focus is unclear, and various standards and numbers are presented in a chaotic format, significantly reducing readability.

The necessity of fast signal processing is not clearly explained. There is no comparison of the processing speed of various methods against time requirements, and the time-domain measurement method is introduced without sufficient justification.

The second research section lists many equipment models from commercial manufacturers, but lacks specific technical details and representative key technologies, which are not closely related to the content before and after this section.

Response : In the introductory section, we will focus on elucidating the pivotal role of electromagnetic compatibility (EMC) and discuss how traditional EMC measurement methods, often necessitating specific facilities and conditions. Consequently, Large-scale platforms are challenged by the increasingly complex electromagnetic environment. This paper will naturally proceed to explore the critical techniques for measuring the electromagnetic radiation characteristics of large equipment or systems in their actual operating environments (in-situ). The manuscript has been meticulously revised to enhance readability, with precise corrections made to various standards and numerical descriptions. Furthermore, in light of the necessity for rapid signal processing, we have conducted a thorough comparative analysis of the processing speeds and temporal requirements of different methodologies in this paper [Page8, Line 240: The measurement speed of FSVR is more than 5 times faster than that of similar spectrometers. Page8, Line 255: The instrument can complete the rapid measurement from 10 Hz to 40 GHz, which is 64000 times faster than other similar products, and has the real-time analysis capability from 325 MHz to 40 GHz.and so on]. Besides, with the trend of technological advancement, researchers' demands for real-time measurement have grown, leading to the gradual emergence of time-domain measurement methods as a new focal point of research.

Round 2

Reviewer 2 Report

Comments and Suggestions for Authors

  1. “The Introduction section spends too much space discussing the importance of EMC and related testing methods, only introducing the issue of large EUT on-site testing in the final paragraph. The focus is unclear, and various standards and numbers are presented in a chaotic format, significantly reducing readability.”

This issue remains unresolved. References [6]-[9] are merely listed without clarifying their specific logical relationships, nor is there further explanation or elaboration. Specifically, what is the connection between the statement "standard laboratory environments are difficult to meet the test requirements" and the preceding text? If the mentioned standards only define testing requirements and large equipment cannot be tested in a standard laboratory, then in-situ testing would serve as a better method to meet these measurement standards. Conversely, if the standards do not apply to large equipment, then in-situ testing would act as an alternative approach outside the scope of those standards. These two lines of reasoning are fundamentally different, and neither conclusion can be directly drawn. This section requires reorganization and clarification of its logic.

  1. “The necessity of fast signal processing is not clearly explained. There is no comparison of the processing speed of various methods against time requirements, and the time-domain measurement method is introduced without sufficient justification.”

Whats missing is not how much faster the new method is compared to the old one, but rather the rationale behind the need for increased processing speed. In other words, what drives the demand for faster processing? Traditional electromagnetic compatibility testing is conducted in laboratories, where speed requirements for the measurement system are typically not a concern. However, when performing in-situ measurements on large equipment, do traditional methods fall short or incur higher economic costs, thereby making fast signal processing essential? As an overview, these points should be clearly addressed in the main text to preempt such questions from readers.

  1. “The second research section lists many equipment models from commercial manufacturers, but lacks specific technical details and representative key technologies, which are not closely related to the content before and after this section.”

The issue in this section persists. The descriptions of the RSA3000B series or the R&S FSVR read more like basic product overviews, especially since DPX is a trademark and lacks technical details. As a result, readers are unable to gain insights into the underlying measurement technology. The authors should provide an explanation, for example, of how Tektronix has implemented its new fast measurement technology to achieve such these performances.

  1. “The resolution of the images is very low, especially Figures 2 and 3, where the text is almost illegible.”

The quality of the images is still the same as before, with no improvement.

Comments on the Quality of English Language

The language quality has improved to some extent, but there are still some issues:

  • On page 1, line 33, "a lot of academic research" should be revised to the plural form.
  • On page 7, lines 197 and 198, "large bandwidth" and "wide bandwidth" are used interchangeably, creating inconsistency.
  • On page 10, line 312, "dB" as a unit should not be italicized.

Author Response

Dear Reviewer 2:

Thank you  for your meticulous work and your helpful comments on our manuscript.  The comments from you and the other reviews were highly insightful and enabled us to greatly improve the quality of our manuscript. The item-by-item responses to each of the comments of the reviews are included. Please see the attachment.

 The literature has been revised carefully according to the comments.We hope that the revision in the manuscript and our accompanying responses will be sufficient to make our manuscript suitable for publication.

However, I am writing to share an urgent matter concerning our project. We are currently facing a project deadline issue. If our paper is not accepted by December, our project will come to an end, and we will be unable to cover the publication fees. This is an external constraint beyond our control, and we regret to inform you of this situation.

Thank you once again for your time and assistance.We look forward to hearing from you at your earliest convenience.  

Yours sincerely,

Zhong-hao Lu
